# Water-Holding Properties of Clinoptilolite/Sodium Polyacrylate-Modified Compacted Clay Cover of Tailing Pond

**DOI:** 10.3390/ijerph192315554

**Published:** 2022-11-23

**Authors:** Xin-Po Sun, Ze-Hao Ding, Yu-Zhang Bi, Xin-Yi Wang

**Affiliations:** 1College of Civil Engineering, Sichuan University of Science & Engineering, Zigong 643000, China; 2Institute of Geotechnical Engineering, Southeast University, Nanjing 210096, China; 3School of Design, Hong Kong Polytechnic University, Hong Kong 999077, China

**Keywords:** clinoptilolite, sodium polyacrylate (Na-PAA), tailing pond, water-holding capacity, soil–water characteristic curve

## Abstract

Clinoptilolite and sodium polyacrylate (Na-PAA) were used as water-retaining agents to improve the water-holding capacity of compacted clay cover (CCC). The optimum moisture content and Atterberg limits of the CCC modified by clinoptilolite and Na-PAA were studied. The soil–water characteristic curve (SWCC) of the CCC modified by clinoptilolite and Na-PAA was studied. The mesostructure of the CCC was analyzed by polarized light microscopy. The test results show that: (1) the optimum moisture content and liquid limit of the CCC modified by clinoptilolite and Na-PAA increased, while the maximum dry density decreased; (2) the SWCC of the CCC modified by clinoptilolite and Na-PAA shifts to the upper right, and the volume moisture content of modified CCC is higher than that of unmodified CCC under the same matrix suction; (3) compared with the unmodified CCC, the air-entry value (AEV) of the clinoptilolite-modified CCC increased by 65.18% at most, and the AEV of the further modified CCC with Na-PAA in-creased by about two times; and (4) the flocculation structure and porosity of modified CCC decreased, and the porosity was distributed uniformly.

## 1. Introduction

Mineral products are the key to and foundation for human civilization’s development. The rise of the mining industry has greatly promoted the development of the smelting and processing industry, the petroleum industry, the electronic industry, the nuclear industry, and the pharmaceutical industry [1]. However, the exploitation and utilization of mineral products are bound to bring harm to the ecological environment around a given mining area, especially with respect to the problem of acid mine drainage (AMD) [2]. The pH of AMD is usually between 2–4, containing a large amount of soluble sulfate, a high concentration of Fe^3+^, and a large amount of Cr, Cu, Mg, Pb, Cd, Zn, and other heavy metals [2,3]. These contaminants seriously threaten the soil and groundwater environment around the mining area. AMD is the product of sulfide (such as pyrite, pyrrhotite, and other sulfate metals) in tailings that are oxidized by air, water, and microorganisms [2]. The production process is shown in Figure 1. So, a barrier cover separating the tailings from the source can be built to control AMD generation from the air and water [4].

Fly ash, geosynthetic clay liner (GCL), compacted clay cover (CCC), sludge, and sawdust are commonly used as the source control methods for AMD pollution problems in engineering [2]. The CCC has the advantages of a wide source, simple construction, and excellent waterproofing and air-blocking performance [5,6], which can effectively inhibit the infiltration of oxygen and rainwater into tailing ponds and prevent the reaction of metal sulfide in the tailings from generating AMD. Due to the influence of water fluctuation and temperature change, the engineering soil (e.g., CCC, roadbed clay, foundation clay, and so on.) is easily cracked and loses its stability, thus causing serious secondary environmental disasters (as shown in Table 1). The width of the CCC’s desiccation cracks can reach 10.0 mm and their depth can be 0.3 m, which can continuously increase to more than 1.0 m if water continues to evaporate and drain away [7,8,9]. Hydraulic conductivity also increases by about 1–4 orders of magnitude due to desiccation cracking [10]. Due to the water/gas migration domination channel [11] formed by desiccation cracking, the oxidation rate and quantity of tailings are increased. Therefore, it is necessary to enhance the water retention performance of CCC to prevent the desiccation cracks caused by the rapid evaporation of water.

Water-retaining agents can quickly absorb and preserve water and reduce desiccation cracking [5]. Soltani et al. [12] show that after adding polyacrylamide (PAM) into the rubber-modified expansive soil, the link between the rubber and soil particles is enhanced due to the role of the polymer binder. It can further inhibit the crack development of rubber-improved soil. Salemi et al. [13] used sodium polyacrylate to study the self-healing ability and anti-wet/dry cycle properties of geosynthetic clay liners. The results show that the self-healing ability of geosynthetic clay liners is improved by the inclusion of a super-absorbent polymer. The wet–dry cycle test results show that the wet–dry cycle resistance of geosynthetic clay liners can be significantly improved by partially replacing bentonite with a super absorbent polymer. In addition, Zhang et al. [14] showed that with the increase in the PAM content, the intersection point and crack length of the compacted saline soil crack network decreased. These results indicate that the increase in PAM reduces the shrinkage strain and defects or pores of saline soil. This proves that PAM can stabilize saline soil under the conditions of a dry–wet cycle. Therefore, using water-retaining agents to modify CCC is an effective technique for preventing CCC from producing desiccation cracks.

The common water-retaining agents can be divided into modified starch, synthetic polymers, modified cellulose, natural substances and their derivatives, blends, and compounds [15]. As an aluminosilicate mineral, zeolite is considered to be a natural inorganic soil conditioner that can improve soil properties such as water retention, acid resistance, adsorption, and cation exchange capacity [16]. As zeolite is a porous medium, its pore grid structure is open to absorbing and storing water. In general, zeolite can reduce soil’s bulk density, increase soil’s total porosity, and increase soil water content [17]. Colombani et al. [18] demonstrated that adding zeolite to silty clay will increase its water-holding capacity, thus limiting the loss of water and solute. He et al. [19] showed that calcareous loess treated with zeolite can increase the water content by 0.4–1.8% under extreme drought conditions, reduce surface runoff, and protect soil from erosion. Du et al. [20,21] added zeolite to a soil–bentonite (SB) vertical barrier wall to explore the influence of zeolite with different particle sizes on hydraulic conductivity. They found that the influence of fine particles (where a mass of soil particles less than 75 μm is more than 50% of the total mass) on the hydraulic conductivity is slight; the coarse zeolite cannot be fully enveloped by bentonite and forms a grid structure through which water can pass, which increases the hydraulic conductivity. Sodium polyacrylate (Na-PAA) is an anionic polyelectrolyte with a negatively charged carboxyl group (COO^−^) in its main chain that binds to water molecules via hydrogen bonds and absorbs hundreds or even thousands of times its weight. The research by Greesling and Schmidhalter [22] shows that when the content of sodium polyacrylate reaches 3 g/L or above, the water-holding capacity is significantly improved. Under dry conditions, Na-PAA can retain moisture in silty clay until the clay is dry, and then gradually releases moisture [13], delaying the total drying time. There are three main reasons why the hydraulic conductivity of sodium polyacrylate–bentonite mixtures can be reduced [13]: (1) the high swelling capacity of sodium polyacrylate reduces the number of pores in the mixture, (2) the water absorption properties of the mixture are stronger than that of bentonite, and (3) the mixtures have a self-healing ability and a resistance to the degradation of their durability caused by the wet–dry cycle.

In this paper, the influence of clinoptilolite and sodium polyacrylate/clinoptilolite composite modifiers on the water-holding capacity of CCC was investigated. The specific experimental concept is shown in Figure 2. The compaction characteristics and liquid limit of the modified CCC with different ratios of clinoptilolite and sodium polyacrylate were investigated. The soil–water characteristic curve (SWCC) of the modified CCC with different ratios of clinoptilolite and sodium polyacrylate was measured by using a filter paper method. The water-holding capacity of the clinoptilolite and sodium polyacrylate-modified CCC were studied to provide some suggestions for the effective use of inorganic and organic water-retaining agents. 

## 2. Materials and Methods

### 2.1. Materials

(1)In situ clay

The test soil was taken from the soil transfer point of a construction site in Sichuan Province (Figure 3); the soil has a reddish-brown hue (Figure 4). After air drying and crushing, the basic physical property indexes of soil were measured as shown in Table 2. The specific gravity of the clay is 2.65, the liquid limit is 35.15%, and the plastic limit is 16.75%. The optimum moisture content of the in situ clay is 15.79%, and the maximum dry density is 1.83 g/cm^3^. Using the NKT6100-D laser particle size analyzer, it was determined that the clay content was 27.55%, the silt content was 65.21%, and the sand content was 7.24%. According to the Unified Soil Classification System [23], the soil was classified as clay with a low plastic limit (CL).

(2)Clinoptilolite

The clinoptilolite used in this experiment is produced in Shijiazhuang, Hebei Province (Figure 5). The basic physical property indexes of the clinoptilolite measured are shown in Table 2. The specific gravity of clinoptilolite is 2.15, its liquid limit is 72.55%, and its plastic limit is 37.8%. Using the NKT6100–D laser particle size analyzer, the clay content was determined to be 23.16%, the silt content was 57.07%, and the sand content was 19.77%. According to the Unified Soil Classification System [23], clinoptilolite was classified as silt with a high liquid limit (MH). In order to explore the influence of clinoptilolite content on soil’s water-holding capacity, and in consideration of the hydraulic conductivity, adsorption performance, gas resistance, and other requirements of CCC, the replacement content of clinoptilolite was set as 0%, 3%, 5%, 10%, and 15% according to the literature [24,25,26].

(3)Sodium polyacrylate

The analytical pure sodium polyacrylate (Na-PAA) used in this experiment was provided by Chengdu Cologne Chemicals Co., Ltd (Chengdu, China; Figure 6). Its molecular formula of (C_3_H_3_NaO_2_)_n_ and a molecular weight of 30–50 million. Its pH (0.1%) is 8.0–9.0, and drying loss is ≤10%. Its CAS is 9003–04–7. According to the content range of super-absorbent polymers given in the literature [17,29,30,31], the content of Na-PAA used in this paper was determined to be 0.35%, which allowed us to avoid wasting manpower and materials to the greatest extent—as mentioned in the preliminary discussion—and achieve the due modification effect.

### 2.2. Sample Preparation

To obtain a uniform CCC sample, the static compaction method [32] was adopted, and the preparation process is shown in Figure 7. The static compaction method is divided into the following steps: (1) evenly mix in situ clay, clinoptilolite, and Na-PAA according to the content in Table 3. (2) Add proper content of deionized water, adjust it to the corresponding water content (i.e., 5.00%, 8.00%, 11.00%, 14.00%, 17.00%, 20.00%, and 23.00% of SWCC test), and seal it with polyethene bag for at least 48 h. (3) Use the static pressure loading device to press the mixed soil into a cylinder of 61.8 mm × 20 mm to ensure the degree of compaction is about 95%. Carry out corresponding tests after curing for 14 days under standard curing conditions (RH ≥ 95%, T = 20 ± 2 °C).

### 2.3. Experiments

#### 2.3.1. Compaction Characteristic

The representative in situ clay was replaced by the quadrant method and mixed with clinoptilolite and Na-PAA according to Table 3. Five portions of clay samples with a moisture gradient of 3% were mixed for 24 h according to the plastic limit values of the samples. The particle size of clinoptilolite and in situ clay used in this paper are much smaller than 20 mm. According to JTG 3430-2020 [28], the light compaction test can be used when the maximum particle size of clay is 20 mm; therefore, the light compaction test was used in this study.

#### 2.3.2. Atterberg Limits 

Atterberg limits constitute an important parameter that is used to reveal the behavior of clay [33]. According to the Liquid and plastic water content joint measurement in JTG 3430-2020 [28], the Atterberg limits of materials in different experimental programs in Table 3 were tested. After passing through a 0.5 mm sieve, the modified in situ clay is mixed with deionized water to reach three humidity states (dry, wet, and intermediate states). In this study, the cone weight of the combined liquid–plastic limit tester is 76 g and the cone angle is 30°. Therefore, the water content corresponding to the 17.0 mm depth of the cone is the liquid limit, and the water content corresponding to the 2.0 mm depth of the cone is the plastic limit.

#### 2.3.3. Filter Paper Method

With reference to the method of Sun et al. [34], matrix suction was measured by using a filter paper method in the moisture absorption process. The CCC is a ring-cutter sample with a diameter of 61.8mm and a height of 20.0 mm. Its degree of compaction is 95.0%, and its moisture content is 5.00%, 8.00%, 11.00%, 14.00%, 17.00%, 20.00%, 23.00%, etc. After the dry density and moisture content were determined, the CCC was prepared by the static compaction method [32].

The steps of the filter paper method are as follows: (1) fully dry Whatman No. 42 filter paper to 0% moisture content. (2) Then, directly attach three pieces of drying filter paper to the bottom of the CCC; the filter paper in the middle shall be the test filter paper, and the other two filter papers shall be used to protect the test filter paper from directly contacting the CCC and being polluted. (3) Put the filter paper and modified CCC into a sealed container (LOCK & LOCK BOX, whose diameter is 114mm, height is 55 mm, and usable volume is about 300 mL) and keep it for 14 days or more at 20 °C and constant humidity [34,35,36] to ensure that the filter paper is wet such that the moisture content is balanced with the CCC’s suction. (4) Finally, after balancing, quickly and accurately measure the moisture content of the filter paper, and calculate the matrix suction using the bilinear calibration curve (Formula (1)) of Leong et al. [35].
(1)$$\begin{array}{l} \mathrm{lgs}=2.909-0.0229w_{f}(w_{f}\geq 47)\\ \mathrm{lgs}=4.945-0.0673w_{f}(26\leq w_{f}\leq 47)\\ \mathrm{lgs}=5.31-0.0879w_{f}(w_{f}<26) \end{array}$$
where s is matrix suction and w*_f_* is the moisture content of filter paper after being balanced.

#### 2.3.4. Polarizing Microscope Test

In this paper, observations were carried out with Shanghai Caikang XPR-3000 polarizing microscope. As the CCCs’ structure is relatively dense and not easy to disperse, the 502 adhesive method proposed by He et al. [37] was used to prepare the thin slices. Specifically, use 502 glue to glue the CCC onto the glass slide, and then use sandpaper to evenly grind it into 30 μm thick, 4–6 cm^2^ large slices. During polishing, the sandpaper position is changed many times to ensure that the thickness of the clay slice is uniform, so that the microstructure of the soil can be observed by a polarizing microscope.

## 3. Results and Analysis

### 3.1. Compaction Characteristics

Figure 8 shows the changes in the maximum dry density and optimum moisture content of the in situ clay under different replacement contents of clinoptilolite and Na-PAA contents. The maximum dry density of Z0P0 is 1.83 g/cm^3^ and its optimum moisture content is 15.79% after fitting the compaction curve with the Gaussian curve. It can be seen from Figure 8f that with the increase in the replacement contents of clinoptilolite and the Na-PAA contents, the maximum dry density of in situ clay decreases, and the optimum moisture content increases. It can be seen from the research results of Qu et al. [38] that the replacement content of clinoptilolite and the content of Na-PAA in this study increase the optimum moisture content of the in situ clay, reflecting the improvement of the water-holding capacity of the in situ clay.

Compared with Z0P0, the optimum moisture content of Z3P0, Z5P0, Z10P0, and Z15P0 increases by 3.29%, 3.86%, 7.16%, and 5.19%, respectively, and the maximum dry density decreases by 0.55%, 2.81%, 3.98% and 4.57%, respectively (as shown in Figure 8a). The optimum moisture content of Z15P0 is only 1.87% lower than that of Z10P0, so it can be determined that the replacement content of clinoptilolite reaches 15%, which has no significant impact on the optimum moisture content of the modified in situ clay. It can be seen from Figure 8b–e that the optimum moisture content of the in situ clay further increases by 5.89%, 7.79%, 9.25%, and 8.68% respectively, with the addition of 0.35% Na-PAA, and its maximum dry density decreases by 2.81%, 3.98%, 4.57%, and 5.78%, respectively. The reason for the decrease in the maximum dry density of the clinoptilolite-modified in situ clay (called mixed soil) may be that the specific gravity of clinoptilolite is smaller than that of in situ clay [17], and the specific gravity of mixed soil decreases, thereby reducing the maximum dry density. In addition, the optimum moisture content of mixed soil shows an increasing trend; one of the reasons for this may be that the special structural characteristics of clinoptilolite [19] can absorb more water. The increase in the replacement content of clinoptilolite reduces the particle size distribution of the mixed soil, effectively increasing the optimal moisture content of the mixed soil [39]. The optimum moisture content and maximum dry density of the mixed soil after adding Na-PAA show the same relationship. The optimum moisture content is higher than that of mixed soil, and the maximum dry density is lower than that of mixed soil. The reason may be that Na-PAA can further reduce the specific gravity of mixed soil. In addition, Na-PAA is a super absorbent polymer [22]; it can absorb more water than clinoptilolite.

### 3.2. Atterberg Limits 

The ability of fine-grained soil to absorb bound water can be directly reflected by the liquid limit or plastic limit (i.e., consistency index) [38]. Therefore, the increase in the liquid limit or plastic limit indicates that the adsorption capacity of soil to water is improved, which further indicates that the water retention capacity of soil is improved.

Figure 9 shows that the liquid limit of Z0P0 is 35.15%, and the liquid limit gradually increases with the increase in the replacement content of clinoptilolite. The liquid limit values of Z3P0, Z5P0, Z10P0, and Z15P0 are 38.60%, 38.80%, 39.90%, and 42.25%, respectively. After being treated with 0.35% Na-PAA, the liquid limit of mixed soil is further improved: 40.00%, 40.60%, 44.25%, and 46.00%, respectively. Similarly, the plastic limit presents the same trend. The reason is that both clinoptilolite and Na-PAA will enable the clay to contain more water due to their good water absorption and retention properties [19,22], which will increase the liquid and plastic limit of in situ clay and its water retention properties [15,38]. It can be seen that clinoptilolite and Na-PAA can improve the water-holding capacity of in situ clay. In addition, Na-PAA further improves the liquid and plastic limits of in situ clay, indicating that the water-holding capacity of Na-PAA is more significant than that of clinoptilolite.

### 3.3. Water-Holding Capacity

#### 3.3.1. Filter Paper Test Results

Figure 10 shows the soil–water characteristic curve (SWCC) of the clinoptilolite-modified CCC with different replacement contents, and Figure 11 shows the soil–water characteristic curve (SWCC) of the Na-PAA–modified mixed soil. Existing research shows that [40], under the same degree of matrix suction, the higher the volume moisture content of CCC, the higher the air−entry value of CCC, while the longer the CCC begins to desaturate, the more the water-holding capacity of soil mass is improved. The shape of the SWCC is shifted to the upper right, so the change in the water-holding capacity can be directly analyzed by examining the shape relationship of the SWCC of CCC modified by different water-retaining agents.

From Figure 10, when the level of matrix suction of Z3P0 is about 50–400 kPa, the SWCC shifts to the upper right compared with Z0P0, but when the matrix suction level is more than 400 kPa, it tends to be consistent with Z0P0. When the matrix suction of Z5P0 is about 50–10,000 kPa, the SWCC shifts to the upper right compared with Z0P0, and when the matrix suction is more than 10,000 kPa, the SWCC tends to be consistent with Z0P0. The SWCC of Z10P0 and Z15P0 have a relatively consistent change trend. When the matrix suction is about 50–50,000 kPa, they are both shifted to the upper right compared with Z0P0, and when the matrix suction is greater than 50,000 kPa, they tend to be consistent with Z0P0. 

According to Figure 11, the SWCC of the Na-PAA–modified mixed soil was entirely shifted to the upper right at 100 kPa–1500 kPa compared with the SWCC of mixed soil. In addition, when the matrix suction is greater than 1500 kPa, the curves tend to be consistent. The results show that 0.35% Na-PAA can further increase the air−entry value (AEV), decelerate the onset of the CCC’s desaturation, and improve the water-holding capacity of the CCC, but the range of the increase is not obvious; therefore, a further increase in Na-PAA content was considered in the later experiment.

The main reasons for the improvement of the water-holding capacity of clinoptilolite and Na-PAA modified CCC are as follows: (1) When the matrix suction of the CCC is less than the AEV, the pore water mainly exists in the form of capillary water and adsorbed water, and the total volume and pore size of the CCC is almost a certain value, which has little effect on the CCC’s moisture loss. (2) When the matrix suction of the CCC reaches the AEV, the air begins to enter the soil particles, and the pore water evaporates in the form of capillary water between the particles. However, the water retained by clinoptilolite [16] and Na-PAA [13] causes the volume moisture content of the improved CCC under the same degree of matrix suction to be higher than Z0P0. (3) The smaller the pore size of the soil, the bigger the matrix potential, the stronger the humidifying effect, and, consequently, the better the water-holding capacity [34]. The clinoptilolite and Na-PAA used in this study will inhibit the increase in the pore size of the CCC [13,21,41,42,43,44], making the pore size smaller than that of Z0P0 (as shown in Figure 12), thus improving the CCC’s water-holding capacity. (4) When the matrix suction increases to the high-suction stage, the pore water of the soil is mainly absorbed water, the inter-particle force remains relatively stable, and residual water content appears. When the replacement content of clinoptilolite is more than 5%, the evaporation of capillary water can continue into the high-suction stage. However, the addition of Na-PAA in the mixed soil cannot further maintain capillary water evaporation at the high-suction stage; this may be due to the low content of Na-PAA, which does not play a role in further improving the CCC’s water-holding capacity.

#### 3.3.2. Fitting of Soil−Water Characteristic Curve

##### Determine the Fitting Model

At present, many soil–water characteristic curve (SWCC) models have been developed [45,46,47], but the SWCC is affected by the soil’s structure, dry density, particle size distribution, stress state, and other factors. A previous study [48] has shown that the influencing factors with respect to the SWCC are various, and the existing SWCC models may not be directly applied or suitable for certain soils. Therefore, this paper conducts a fitting analysis of several common models and compares the reliability of the models through the sum of the residual squares (SSR) [49,50].
(2)$$\mathrm{SSR}=\sum_{i=1}^{n}{w_{i}(\theta_{wi}-\theta_{ci})}^{2}$$
where $w_{i}$ = weighting factor, which is equal to 1.0 [51]; $\theta_{wi}$ = measured moisture content at a certain pressure level; and $\theta_{ci}$ = calculated moisture content from each model at the same pressure level.

In order to further quantitatively analyze the influence of clinoptilolite and Na-PAA on the SWCC of the CCC, the Lsqcurve function built in MATLAB was used to fit the SWCC. Four SWCC models were considered during fitting, namely, the two Fredlund and Xing models [45], Van Genuchten model [46], and the Gardner model [47]. The specific expressions and parameter meanings of each model are shown in Table 4. Among them, the $\varphi_{re}$ in FX1 is not the actual residual suction value. Fredlund et al. [45] mentioned that in most cases, $\varphi_{re}$ can be between 1500 kPa and 3000 kPa, and well-fitting results can be obtained for the soil–water characteristic curves of different types of soil. When the residual suction value is difficult to determine, it is recommended that be taken as 3000 kPa [52]. For the convenience of calculation, 3000 kPa was employed in this paper.

Table 5 shows the SSR of different SWCC models. As can be seen from Table 5, Z0P0, Z10P0, Z15P0 and Z15P0.35 had the smallest SSR when fitted using the VG model, whereas other samples had the smallest SSR when fitted using the FX and FX1 models. However, the difference between the SRR of the VG model and the FX and FX1 models are very small. In addition, only one SSR of VG model is not less than 1 × 10^−3^, indicating that the models provide an appropriate and acceptable fit to the measured data [51,53], so the FX1 model’s fitting results were selected for analysis.

##### Effect of Clinoptilolite and Na-PAA on Air−Entry Value (AEV) of CCC

Table 6 shows the results of the FX1 model’s fitting parameters for different mixed soils, mainly analyzing the change in the matrix suction air entry value (AEV). The AEV of Z0P0 is 340.13 kPa. With the increase in the replacement content of clinoptilolite, the AEV shows an increasing trend. The AEV of Z15P0 is 549.14 kPa, which increased by 45.92%. Compared with Z0P0, the AEV of Z3P0, Z5P0, Z10P0, and Z15P0 were increased by 15.09%, 42.73%, 51.13% and 65.18%, respectively.

After the mixed soil was treated with Na-PAA, the AEV reached about double that of each mixed soil. The higher the AEV, the higher the desaturation suction value [40,45,51], and the higher the water-holding capacity of the mixed soil, which shows that clinoptilolite and the Na-PAA modified CCC have a good water-holding capacity, and the water-holding capacity of Na-PAA is better than clinoptilolite. As mentioned above [13,16,41], clinoptilolite and Na-PAA can reduce the pore size of the soil mass, and clinoptilolite and Na-PAA can retain additional water to compensate the water lost by the CCC itself. Therefore, the AEV of the modified CCC increases and its water-holding capacity is improved.

### 3.4. Mesostructure

In this study, the effect of the CCCs’ mesostructures on the water retention capacity of the clinoptilolite and Na-PAA–modified CCC was investigated by a polarizing microscope. The analysis diagram of the polarizing microscope is shown in Figure 12, wherein Figure 12a is the mesostructure of the CCC and Figure 12b is the mesostructure of the clinoptilolite/Na-PAA-modified CCC. 

It can be seen from the pictures that the CCC particles are a flocculation structure, resulting in large and many pores (the pores are occupied by 502 glue), while the particles of the clinoptilolite/Na-PAA modified CCC are relatively dispersed, which may be because Na-PAA swells and forms a gel that clogs the clay pores [13,43,44], or because Na-PAA has strong viscosity after absorbing water, thus forming a “glue”-like effect between the clay particles. This may be because, for the unmodified CCC, the interaction between the soil particles is dominant, and the low viscosity of the pore fluid will not inhibit the interaction between the particles, so a flocculation structure will be formed. As the viscosity of the pore fluid increases, hydrophilic and charged hydrogels are formed. The higher the hydrophilic gel, the better the water-holding capacity of the soil [54]. In addition, the clinoptilolite used in this paper is a fine particle [20,21], which can fill pores as much as possible, so the pore size is reduced. Similarly, some researchers obtained similar conclusions [55]. They believe that the flocculation structure of the unmodified CCC will form larger pores between particle aggregates, leading to the loss of water in the sample. After adding a biopolymer, the sample will form a dispersed structure, making the particles closely connected, thus reducing the loss of water and increasing the water-holding capacity. Other results [41] also show that the water-holding capacity of soil is affected by the size of pores. Due to the existence of macropores, the soil will begin to drain under low-level matrix suction, so the soil’s air entry value is low.

## 4. Discussion

### 4.1. Analysis of the Influence of Different Water-Retaining Agents on Optimum Moisture Content and Liquid Limit

Previous studies have shown that [15,38] the greater the optimum moisture content and liquid limit, the more water the soil can absorb and the greater the water-holding capacity. Therefore, in order to compare the influence of different water-retaining agents on the optimum moisture content, a dimensionless parameter *w_omcd_* (*w_omcd_* = *w_omci_*/*w_omc0_*, where *w_omci_* is the optimum moisture content of the soil mass modified by different water-retaining agent materials, and *w_omc0_* is the optimum moisture content of soil mass before modification) is proposed. Similarly, in order to compare the effects of different water-retaining agents on the liquid limit value, another dimensionless parameter *W_Ld_* (*W_Ld_* = *W_Li_*/*W_L0_*, where *W_Li_* is the liquid limit value of the soil mass modified by different water-retaining agent materials, and *W_L0_* is the liquid limit value of the soil mass before modification) is proposed. Figure 13a shows the influence of different water-retaining agents on the optimum moisture content of the soils. The water-retaining agents include clinoptilolite, Na-PAA, straw ash, polyacrylamide, and a biomass polymer, and the soil types include low-plasticity clay, seashore saline soil, and high-plasticity clay. Figure 13b shows the influence of the different water-retaining agents on the soils’ liquid limit. The water-retaining agents include clinoptilolite, Na-PAA, straw ash, polyacrylamide, a biomass polymer, attapulgite, and diatomaceous earth, and the soil types include low-plasticity clay, high-plasticity clay, mucky loam, and seashore saline soil.

As can be seen from Figure 13a, *w_omcd_* shows a less pronounced decreasing trend with an increasing number of biomass polymers [56]. The *w_omcd_* of the soil mass modified by clinoptilolite, straw ash [38], polyacrylamide [57], and Na-PAA showed an increasing trend, indicating that the soil mass modified by these water-retaining agents had a good water-holding capacity. The clinoptilolite and Na-PAA used in this paper have better *w_omcd_* performance than other water-retaining agents. From Figure 13b, it can be seen that the *W_Ld_* of soil increases with the addition of water-retaining agents, while diatomaceous earth has a certain inhibiting effect on *W_Ld_* [15]; this may be because diatomaceous earth and soil’s water absorption capacity is the same, so the *w_omcd_* did not significantly improve. Biopolymers [56] have the highest *W_Ld_*-lifting capacity, showing an opposite behavior to that of *w_omcd_*, because biopolymers increase the pore fluid viscosity of soils, and the clay–polymer link network is formed by a cationic bridge and a hydrogen bond, so the *W_Ld_* is increased. Although the effect of the Na-PAA used in this paper on the increase in the soil liquid limit is not as strong as that of locust bean gum in the literature [56], the economic benefits of the Na-PAA used in this paper are lower than those of locust bean gum (as shown in Table 7). The clinoptilolite and Na-PAA used in this paper also have good *W_Ld_* enhancement effects and are better than biochar [58], attapulgite [15], and straw ash [38]. Therefore, clinoptilolite and Na-PAA have a good enhancing effect on *W_Ld_* and *w_omcd_*, and show that the CCC modified by both of these materials has good water-holding capacity and is highly economical.

### 4.2. Analysis of the Influence of Different Water-Retaining Agents on the AEV

The fitting parameter of the VG model represents the AEV. When the AEV is larger, the suction of the soil mass to initiate desaturation is greater, and its water-holding capacity is stronger; on the contrary, its water-holding capacity is weaker [45]. In order to compare the influence of different water-retaining agent materials on the AEV of soil mass, the dimensionless parameter *AEV_d_* (*AEV_d_* = *AEV_i_*/*AEV_0_*, where *AEV_i_* is the AEV after the improvement of different water-retaining agents, and *AEV_0_* is the AEV of the soil mass before treatment) is used for analysis. Figure 14 shows the influence of different water-retaining agents on the AEV. The water-retaining agents include clinoptilolite, Na-PAA, xanthan gum, gellan gum, guar gum, fly ash, wheat straw, wheat husk, and biochar. The soil types include silty clay, low-plasticity clay, loam, salt-stagnant soil, and chernozem.

It can be seen from Figure 14 that the *AEV_d_* increases with the increase in the clinoptilolite content, and the increase in AEV in this paper is basically consistent with the results in the literature [59]. In this study, the *AEV_d_* tended to increase with the content of Na-PAA, while in the literature [55], the *AEV_d_* decreased after a modification with Xanthan gum and gellan gum, while *AEV_d_* increased after a modification with guar gum. The porosity of soil after treatment with xanthan gum and gellan gum may be larger than that before treatment, but guar gum increases the *AEV_d_* due to its adhesiveness [56]. However, the content of the cross-linked polymer Na-PAA in this paper is only 0.35%, so it can be surmised that Na-PAA has better water-holding capacity than guar gum. In the literature [60], the effect of fly ash on the *AEV_d_* of low-liquid limit clay is higher than that of clinoptilolite (or Na-PAA) in this paper. This may be because the particle size of fly ash is smaller than that of the clinoptilolite in this paper [60], allowing the fly ash to fill the soil pores and change the pore size distribution of the soil. The application of wheat straw and wheat husks in the literature [61] has a lower effect on the *AEV_d_* than that of clinoptilolite (or Na-PAA) in this paper. Similarly, the effect of biochar on the water-holding capacity of salt-stagnant soil in [62] is lower than that of clinoptilolite (or Na-PAA) in this paper. It can be concluded that the CCC modified with clinoptilolite and clinoptilolite/Na-PAA can enhance the *AEV_d_*, and it is an effective method to improve the water-holding capacity of the soil.

## 5. Conclusions

This paper aimed at the engineering problem concerning a water/gas migration dominant channel formed by the desiccation cracking of compacted clay cover in tailing ponds. Clinoptilolite and Na-PAA were used as water-retaining agents to enhance CCC’s water-holding capacity. The water-holding capacity of clinoptilolite/Na-PAA–modified CCC was studied by a compaction test, Atterberg limits test and matrix suction measurements were determined by a filter paper method, and the mesostructure of the CCC was analyzed by a polarizing microscope. Based on the results, the following conclusions can be drawn:(1)With the increase in the replacement content of clinoptilolite and the Na-PAA content, the maximum dry density of the modified CCC decreases, the optimum moisture content increases, and the liquid limit increases. In addition, the optimum moisture content and the liquid limit value of the clinoptilolite-modified CCC are further improved after being treated with Na-PAA. The maximum increases in the optimum moisture content and the liquid limit are 9.25% and 30.87%, respectively.(2)Clinoptilolite and Na-PAA have a good effect on improving the AEV of the CCC matrix. When the replacement content of clinoptilolite is 15%, the AEV is 65.18% higher than Z0P0, and Na-PAA can further increase the AEV to 1065.62 kPa at most, indicating that both clinoptilolite and Na-PAA can improve the water-holding capacity of CCC, and Na-PAA has an even more excellent enhancing effect.(3)The Clinoptilolite and Na-PAA-modified CCC had a decreased number of flocculation structures and the CCC particles were distributed uniformly, which reduced the pore diameters, reduced the water evaporation pathways, and enhanced the water-holding capacity of the CCC.

Although the water-retaining agents used in this study can effectively improve the water-holding capacity of CCC, the effect of sodium polyacrylate (Na-PAA) on the water-holding capacity of CCCs in different amounts was not considered, and the effect of clinoptilolite and Na-PAA on the “Dominant Channel”, hydraulic conductivity, and gas resistance of CCC were also not considered. In the future, the corresponding research will be carried out for desiccation cracks, impermeability, and gas resistance. Further studies are warranted to explore these issues in the future.

## Figures and Tables

**Figure 1 ijerph-19-15554-f001:**
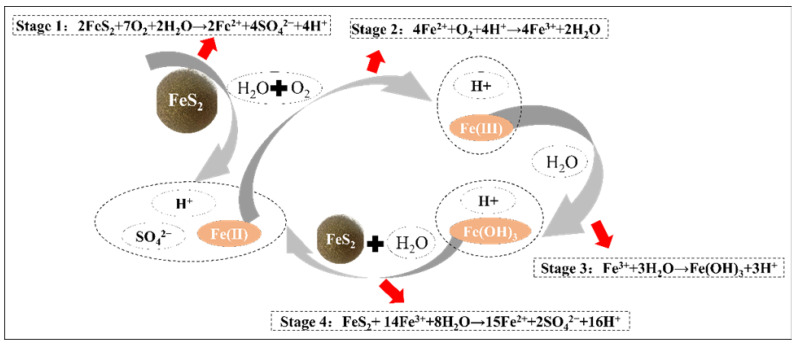
The schematic diagram of acid mine drainage (AMD).

**Figure 2 ijerph-19-15554-f002:**
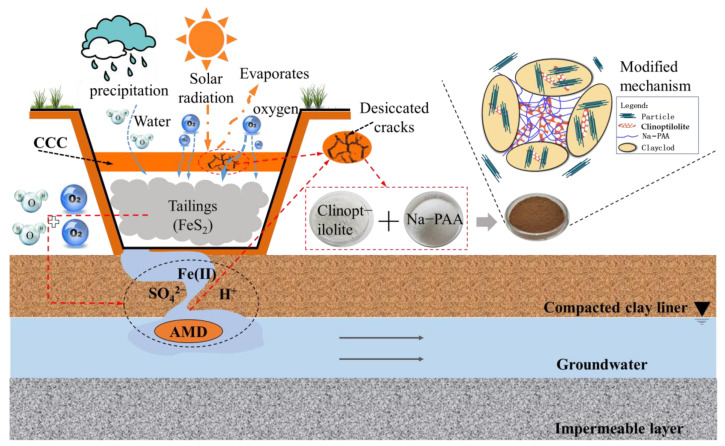
Problems to be solved and research approach.

**Figure 3 ijerph-19-15554-f003:**
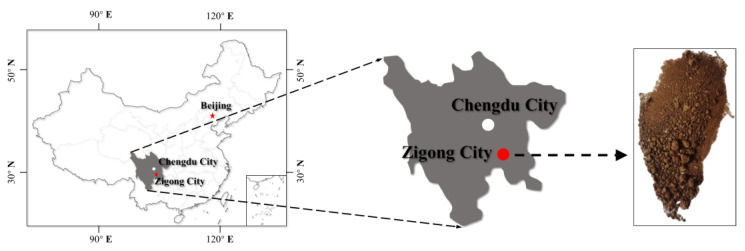
Sampling site for in situ clay.

**Figure 4 ijerph-19-15554-f004:**
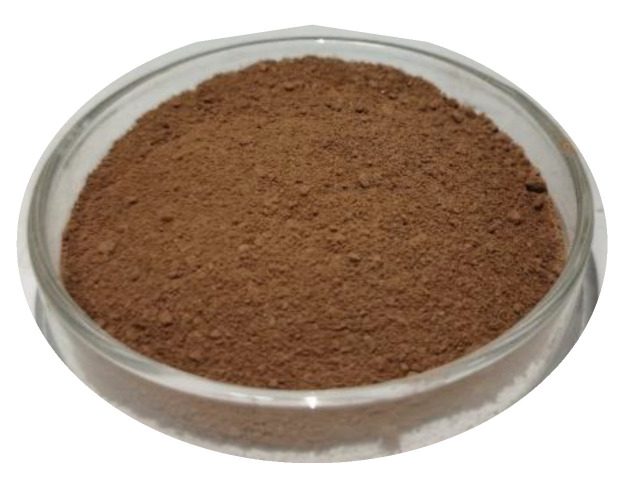
Picture of in−situ clay.

**Figure 5 ijerph-19-15554-f005:**
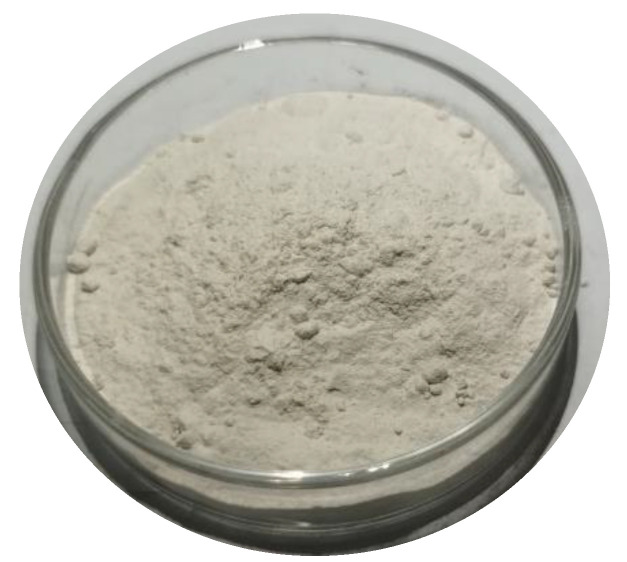
Picture of clinoptilolite.

**Figure 6 ijerph-19-15554-f006:**
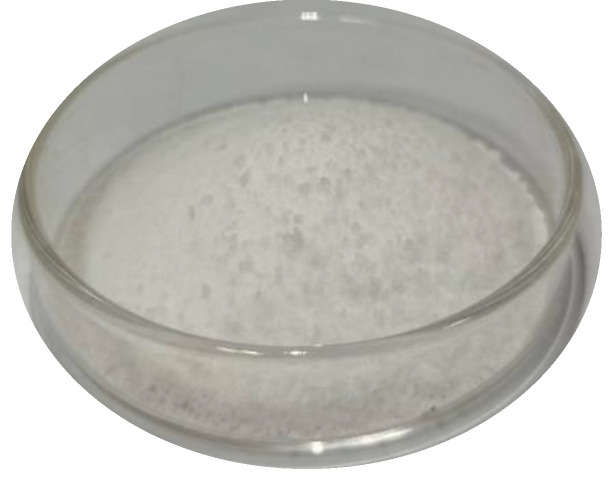
Picture of sodium polyacrylate.

**Figure 7 ijerph-19-15554-f007:**
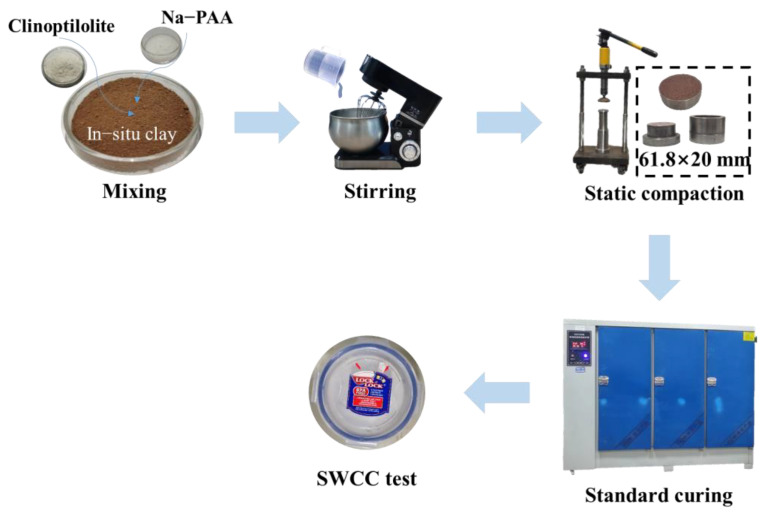
Schematic illustration for the preparation of CCC.

**Figure 8 ijerph-19-15554-f008:**
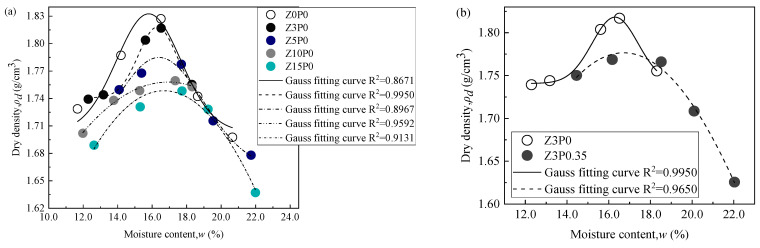
Compaction curve of CCC modified by different water-retaining agent materials. (**a**) Different replacement contents of clinoptilolite; (**b**) Z3P0.35, (**c**) Z5P0.35, (**d**) Z10P0.35, and (**e**) Z15P0.35; (**f**) optimum moisture content and maximum dry density changes.

**Figure 9 ijerph-19-15554-f009:**
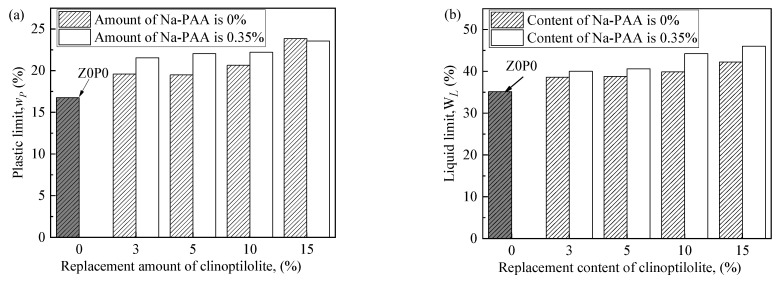
The Atterberg limits. (**a**) Plastic limit; (**b**) Liquid limit.

**Figure 10 ijerph-19-15554-f010:**
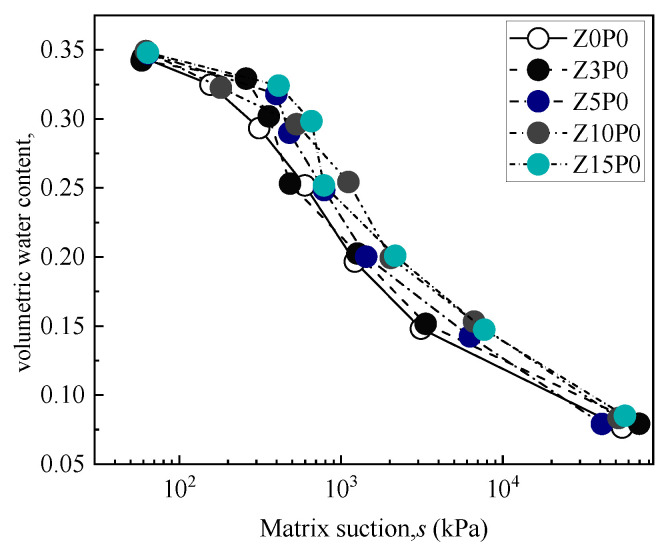
The soil–water characteristic curve of modified CCC with different replacement contents of clinoptilolite.

**Figure 11 ijerph-19-15554-f011:**
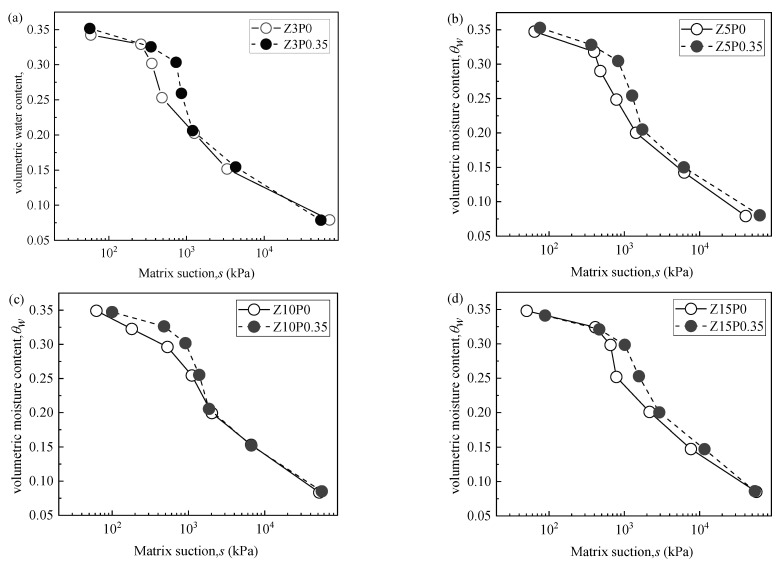
The soil–water characteristic curve of Na-PAA modified mixed soil. (**a**) SWCC of Z3P0 and Z3P0.35; (**b**) SWCC of Z5P0 and Z5P0.35; (**c**) SWCC of Z10P0 and Z10P0.35; (**d**) SWCC of Z15P0 and Z15P0.35.

**Figure 12 ijerph-19-15554-f012:**
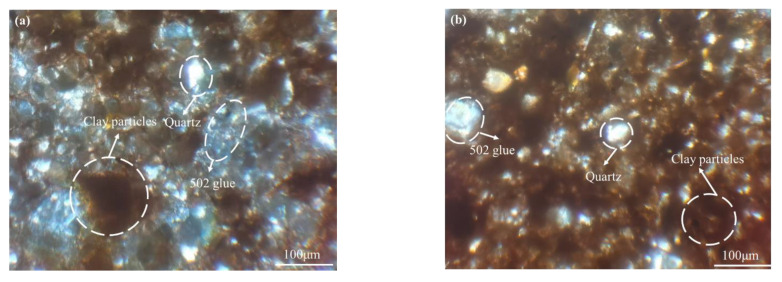
Polarization microscopy analysis diagrams (orthogonal polarization): (**a**) CCC and (**b**) clinoptilolite/Na PAA modified CCC.

**Figure 13 ijerph-19-15554-f013:**
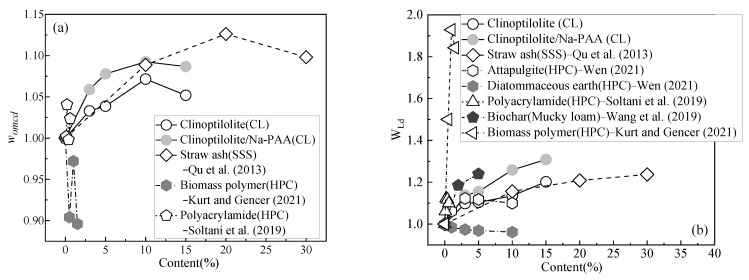
The influence of different water-retaining agents materials on the optimum moisture content and liquid limit. (**a**) Optimum moisture content *w_omcd_* [38,56,57], (**b**) liquid limit W*_Ld_* [15,38,56,57,58].

**Figure 14 ijerph-19-15554-f014:**
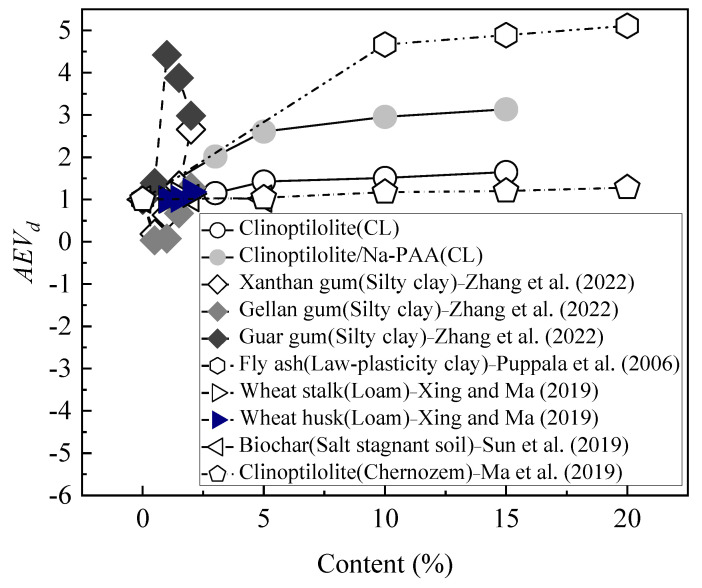
Influence of different water-retaining agents on air−entry value (*AEV_d_*) [55,59,60,61,62].

**Table 1 ijerph-19-15554-t001:** Environmental disasters caused by soil cracking or instability.

No.	Date (yr)	Location	Damage	Pictures
1	2010	Southwest China	Severe drought caused soil cracking of 1200 dams in Chongqing	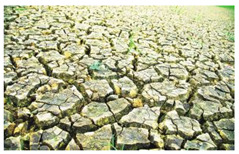
2	2022	Laixi, China	Gullies and cracks formed in the dam body of the tailings pond, causing no harm	Lack of information
3	2007	Yueyang, China	The mountain on the left bank of the tailings pond has many cracks, threatening the safe operation of the tailings pond	Lack of information
4	2008	Shanxi, China	The tailings pond was liquefied in a large area and the dam body became unstable, resulting in 277 deaths, 4 missing people, and 33 injured.	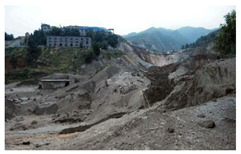
5	2022	Shanxi, China	Collapse caused by leakage and instability of tailings pond dam	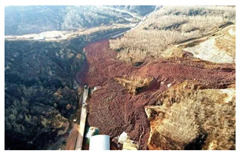
6	2000	the Philippines	218 dead and 100 missing	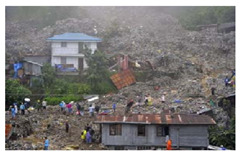
7	2005	Indonesia	61 people dead, 90 people missing, and 75 houses were destroyed	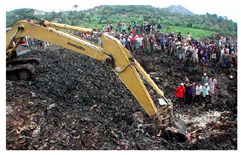

**Table 2 ijerph-19-15554-t002:** The basic physical property indexes of the in situ clay and clinoptilolite.

Soil Properties	In Situ Clay	Clinoptilolite	Methods
Specific gravity, *G_s_*	2.65	2.15	ASTM D5550 [27]
Atterberg limits			JTG 3430-2020 [28]
Liquid limit, *W_L_* (%)	35.15	72.55	
Plastic limit, *W_P_* (%)	16.75	37.80	
Plasticity index, *I_P_* (%)	19.40	34.75	
Light compaction test			JTG 3430-2020 [28]
Optimum moisture content, *w_omc_* (%)	15.79	-	
Maximum dry density, *ρ_dmax_* (g/cm^3^)	1.83	-	
Particle composition			NKT6100-D laser particle size analyzer
Sand content (>75 μm) (%)	7.24	19.77	
Silt content (5–75 μm) (%)	65.21	57.07	
Clay content (<5 μm) (%)	27.55	23.16	
USCS classification	CL	MH	ASTM D2487 [23]

-: Indicates that the parameters have not been tested.

**Table 3 ijerph-19-15554-t003:** Details of the experimental program.

Group No.	Replacement Content of Clinoptilolite (G)/%	Content of Na-PAA (P)/%
Z0P0	0	/
Z3P0	3	
Z5P0	5	
Z10P0	10	
Z15P0	15	
Z3P0.35	3	0.35
Z5P0.35	5	
Z10P0.35	10	
Z15P0.35	15	

Note: ‘G’ is the mass ratio of clinoptilolite to clay. ‘P’ represents the percentage of the total mass. In ZxPy, x represents the G, and y represents the P.

**Table 4 ijerph-19-15554-t004:** Soil–water characteristic curve models [45,46,47].

Models	Equation	Parameters
Van Genuchten model (VG)	$\theta_{w}=\frac{\theta_{s}-\theta_{r}}{{\left[1+{\left(\frac{s}{a}\right)}^{n}\right]}^{m}}+\theta_{r}$	$\theta_{w}$: Volumetric moisture content $\theta_{r}$ : Residual moisture content $\theta_{s}$: Saturated moisture content s: Matrix suction, kPa a, n, m: Fitting parameters, a: related to air-entry value (AEV), kPa, m = 1 − 1/n
Fredlund and Xing model (FX)	$\theta_{w}=\frac{\theta_{s}-\theta_{r}}{{\left\{\ln\left[e+{\left(\frac{s}{a}\right)}^{n}\right]\right\}}^{m}}+\theta_{r}$	
Fredlund and Xing model (FX1)	θw=θs{ln[e+(sa)n]}m·C(s)C(s)=1−ln(1+sφre)ln(1+106φre),φre=3000 kPa	$\varphi_{re}$: Suction corresponding to the residual moisture content, kPa. The rest of the parameters match those given above
Gardner model (GD)	$\theta_{w}=\frac{\theta_{s}-\theta_{r}}{1+{\left(\frac{s}{a}\right)}^{n}}+\theta_{r}$	Parameter meanings match those given above

**Table 5 ijerph-19-15554-t005:** Sum of squares of fitting residuals of different SWCC models.

Models	Group No.								
	Z0P0	Z3P0	Z5P0	Z10P0	Z15P0	Z3P0.35	Z5P0.35	Z10P0.35	Z15P0.35
VG	1.5 × 10^−5 #^	8.4 × 10^−4^	6.3 × 10^−4^	3.1 × 10^−4 #^	9.2 × 10^−4 #^	1.7 × 10^−3^	8.8 × 10^−4^	8.7 × 10^−4^	5.2 × 10^−4 #^
FX	3.9 × 10^−4^	8.3 × 10^−4 #^	5.5 × 10^−4^	8.0 × 10^−4^	1.0 × 10^−3^	1.3 × 10^−3 #^	6.6 × 10^−4 #^	6.0 × 10^−4 #^	5.3 × 10^−4^
FX1	3.0 × 10^−4^	1.3 × 10^−3^	5.1 × 10^−4#^	3.7 × 10^−4^	1.0 × 10^−3^	1.9 × 10^−3^	1.2 × 10^−3^	5.6 × 10^−3^	4.1 × 10^−3^
GD	7.4 × 10^−5^	1.1 × 10^−3^	9.9 × 10^−4^	4.3 × 10^−4^	1.2 × 10^−3^	2.0 × 10^−3^	1.2 × 10^−3^	1.2 × 10^−3^	7.8 × 10^−4^

Note: ^#^ represents the minimum of the SSR.

**Table 6 ijerph-19-15554-t006:** Fitting parameters.

PARM.	Group No.								
	Z0P0	Z3P0	Z5P0	Z10P0	Z15P0	Z3P0.35	Z5P0.35	Z10P0.35	Z15P0.35
a(AEV)	340.13	391.46	485.47	514.03	561.84	686.55	886.74	1005.59	1065.62
n	1.53	1.60	1.59	1.39	1.51	1.78	1.75	1.81	1.56
*θ_r_*	0.06	0.07	0.06	0.03	0.06	0.07	0.07	0.08	0.06
*θ_s_*	0.35	0.35	0.36	0.35	0.36	0.36	0.36	0.35	0.35

**Table 7 ijerph-19-15554-t007:** Comparison of economic benefits of partial water-retaining agents.

No.	Materials	Purity Specification	Price (/100 g)	Contents (Refer to Dry Soil)	Mass of Dry Soil	Cost
1	Locust bean gum	Food-grade	$2.8	e.g., 1%	e.g., 10 kg	$2.800
2	Straw ash	-	$0.14	e.g., 20%		$2.800
3	Polyacrylamide	AR	$1.66	e.g., 0.6%		$0.996
4	Attapulgite	-	$0.83	e.g., 10%		$8.3
5	Biochar	Rice straw	$0.28	e.g., 15%		$4.2
6	clinoptilolite	-	$0.23	15%		$3.45
7	Na-PAA	AR	$1.38	0.35%		$0.483

## Data Availability

The data that support the findings of this study are available from the first author, Xin-Po Sun, upon reasonable request.

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
