# Peer review of "Water-Holding Properties of Clinoptilolite/Sodium Polyacrylate-Modified Compacted Clay Cover of Tailing Pond"

_ijerph, 2022, doi:10.3390/ijerph192315554_

Round 1

Reviewer 1 Report

Please add some photos of the materials and the experimentation setup. Overall, the paper is well written and adequate discussion is carried out on this topic. It can be accepted after minor revision.

Author Response

Reviewer #1: Please add some photos of the materials and the experimentation setup. Overall, the paper is well written and adequate discussion is carried out on this topic. It can be accepted after minor revision.

Response to Reviewer #1:

Above all, we would like to thank you for the specific and constructive comments on this manuscript. It has helped shape the work in all aspects and allowed us a useful examination of our work and the improvements that have been made. Below are specific responses to the comments and criticisms.

Reviewer #1: Please add some photos of the materials and the experimentation setup.

Response: Thank you for your advice. The materials photos and experimentation setup have been added in the manuscript. As follows:

Figure 4. picture of in-situ clay

Figure 5. picture of clinoptilolite

Figure 6. picture of sodium polyacrylate

2.2 Sample prepartion

To obtain a uniform CCC sample, the static compaction method[20] was adopted, and the preparation process is shown in Figure 7. The static compaction method is divided into the following steps: (1) Evenly mix in-situ clay, clinoptilolite and Na-PAA according to the content in Table 3. (2) Add proper content of deionized water, adjust it to the corresponding water content (i.e. 5.00%, 8.00%, 11.00%, 14.00%, 17.00%, 20.00%, 23.00% of SWCC test), and seal it with polyethylene bag for at least 48 h. (3) The static pressure loading device is used to press the mixed soil into a cylinder of 61.8mm * 20mm to ensure the degree of compaction is about 95%. Carry out corresponding tests after curing for 14 days under standard curing conditions (RH ≥ 95%, T=20 ± 2 ℃).

Figure 7. Schematic illustration for the preparation of CCC

Reviewer 2 Report

General comments:

In this paper, liquid limit value, optimum water content, and SWCC are used to illustrate the improvement effect of clinoptilolite and sodium polyacrylate on the water retention performance of silty clay. Some conclusions were presented in the paper. The material and method for improving the water retention property of silty clay are of certain significance to academic and engineering practice. I think this paper needs major revision.

Nonetheless, some issues in this paper need to be modified and explained by the authors.

Specific comments:

1. Line 39-40: The boxes for each step in Figure 1 should be adjusted to be clear.

2. Line 56-58: It should be explained why clinoptilolite can improve soil properties such as water retention.

3. Line 78-79: Can self-healing ability and the ability to withstand dry and wet cycles be classified as the third reason? If not, please quote relevant literature.

4. Line 80-82: “This paper studied the water-holding capacity of clinoptilolite, sodium polyacrylate and silty clay mixtures.” and “The influence of clinoptilolite and sodium polyacrylate/clinoptilolite composite modifiers on the water-holding capacity of silty clay was investigated.” Are these two sentences repeated?

5. Line104-109: Why choose clinoptilolite of this size?

5. Line 94-111: Test methods or specifications describing relevant physical parameters.

6. Line 123-125: Relevant specifications shall be quoted.

7. Line 127-131: Reduce the text to facilitate reading.

8. Line 134: Relevant specifications shall be quoted.

9. Line 204-209: Reduce the text to facilitate reading.

10. Line 306: Check the spelling of Z5P0.

11. about the title: it should be related to the mine-contaminated site.

12. line 25-89: the English need to be improved. The purpose of your study is CCC modification, the relevant research should be added to this part.

13. line 94: you should give an explanation about silty clay, this is the materials that you used in CCC?

14. line 124 and line 134: add the ASTM standards.

15. line 153: what about the size of the LOCK size? What is the humidity in your experiments? Does it reasonably for your experiments?

16. line 153: Is keeping it for 14 days enough?

17. line 333: the discussion part is not enough, what about the micromechanism? 

Author Response

Reviewer #2: In this paper, liquid limit value, optimum water content, and SWCC are used to illustrate the improvement effect of clinoptilolite and sodium polyacrylate on the water retention performance of silty clay. Some conclusions were presented in the paper. The material and method for improving the water retention property of silty clay are of certain significance to academic and engineering practice. I think this paper needs major revision.

Nonetheless, some issues in this paper need to be modified and explained by the authors.

Response to Reviewer #2:

We would like to thank Reviewer 2 for his patience and diligence in reviewing this manuscript. The comments are essential and have helped us improve the manuscript drastically. Please find below responses to specific comments and criticisms raised:

Reviewer #2: Line 39-40: The boxes for each step in Figure 1 should be adjusted to be clear.

Response: Thank you for your advice. Figure 1 has been modified in the manuscript, which is as follows:

Figure 1. The schematic diagram of Acid mine drainage(AMD).

Reviewer #2: Line 56-58: It should be explained why clinoptilolite can improve soil properties such as water retention.

Response: Thank you for your comment. The reason why clinoptilolite can improve soil properties such as water retention has been added to the manuscript. It is as follows:

As clinoptilolite is a porous medium, its pore grid structure is open to absorb and store water. In general, zeolite can reduce soil bulk density, increase soil total porosity and increase soil water content[22].

Reviewer #2: Line 78-79: Can self-healing ability and the ability to withstand dry and wet cycles be classified as the third reason? If not, please quote relevant literature.

Response: Thank you for your comment. After checking, it has been listed as the third point, which is as follows:

There are two main reasons why the hydraulic conductivity of sodium polyacrylate-bentonite mixtures can be reduced[18]: (1) the high swelling capacity of sodium polyacrylate reduces the number of pores in the mixture; (2) the water absorption properties of the mixture is stronger than that of bentonite; (3) the mixtures have the self-healing ability and the resistance to the degradation of durability caused by the wet-dry cycle.

Reviewer #2: Line 80-82: “This paper studied the water-holding capacity of clinoptilolite, sodium polyacrylate and silty clay mixtures.” and “The influence of clinoptilolite and sodium polyacrylate/clinoptilolite composite modifiers on the water-holding capacity of silty clay was investigated.” Are these two sentences repeated?

Response: Thank you for your advice. These two sentences have been modified. As follows:

In this paper, the influence of clinoptilolite and sodium polyacrylate/clinoptilolite composite modifiers on the water-holding capacity of CCC was investigated.

Reviewer #2: Line 104-109: Why choose clinoptilolite of this size?

Response: Thank you for your comment. The reasons for selecting clinoptilolite of this size are: (1) If large particles of clinoptilolite are selected, the particle size distribution of soil will become larger, which is not conducive to promoting the water holding performance[23]. (2) It has been shown that when the particle size<75% accounts for more than 50%, it has good anti-seepage performance and compression performance[15-16].

Reviewer #2: Line 94-111: Test methods or specifications describing relevant physical parameters.

Response: Thank you for your advice. Relevant test methods and specifications have been added in Table 1. As follows:

Table 1. The basic physical property indexes of the in-situ clay and clinoptilolite.

Soil properties

In-situ clay

Clinoptilolite

Methods

Specific gravity, Gs

2.65

2.15

ASTM D5550[47]

Atterberg limits

JTG 3430-2020[48]

Liquid limit, WL(%)

35.15

72.55

Plastic limit, WP (%)

16.75

37.80

Plasticity index, IP(%)

19.40

34.75

Standard compaction test

JTG 3430-2020[48]

Optimum moisture content, wopt (%)

15.81

-

Maximum dry density, ρdmax (g/cm3)

1.83

-

Particle composition

NKT6100-D laser particle size analyzer

Sand content(>75μm)(%)

7.24

19.77

Silt content(5-75μm)(%)

65.21

57.07

Clay content(<5μm)(%)

27.55

23.16

USCS classification

CL

MH

ASTM D2487[49]

-: Indicates that the parameters have not been tested.

Reviewer #2: Line 123-125: Relevant specifications shall be quoted.

Response: Thank you for your advice. It has been modified in the manuscript.

Reviewer #2: Line 127-131: Reduce the text to facilitate reading.

Response: Thank you for your advice. It has been modified in the manuscript.

Reviewer #2: Line 134: Relevant specifications shall be quoted.

Response: Thank you for your advice. It has been modified in the manuscript.

Reviewer #2: Line 204-209: Reduce the text to facilitate reading.

Response: Thank you for your advice. It has been modified in the manuscript.

Reviewer #2: Line 306: Check the spelling of Z5P0.

Response: Thank you for your advice. It has been modified in the manuscript.

Reviewer #2: about the title: it should be related to the mine-contaminated site.

Response: Thank you for your advice. The title has been changed to “water-holding properties of clinoptilolite/sodium polyacrylate modified compacted clay cover of tailing pond”

Reviewer #2: line 25-89: the English need to be improved. The purpose of your study is CCC modification, the relevant research should be added to this part.

Response: Thank you for your advice. We apologize for the poor language of our manuscript. We worked on the manuscript for a long time and the repeated addition and removal of sentences and sections obviously led to poor readability. We have now worked on both language and readability. The modification of CCC has been added in the manuscript, which is as follows:

Water-retaining agents can quickly absorb and preserve water and reduce desiccation cracking [5]. Soltani et al. [62] show that after adding polyacrylamide(PAM) into the rubber-modified expansive soil, the link between rubber and soil particles is enhanced due to the role of polymer binder. It can further inhibit the crack development of rubber-improved soil. Salemi et al. [18] used sodium polyacrylate to study the self-healing ability and anti-wet/dry cycle ability of geosynthetic clay liners. The results show that the self-healing ability of geosynthetic clay liners is improved by super-absorbent polymer inclusion. The wet/dry cycle test results show that the wet/dry cycle resistance of geosynthetic clay liners can be significantly improved by partially replacing bentonite with a super absorbent polymer. And Zhang et al. [63] showed that with the increase in PAM content, the intersection point and crack length of the compacted saline soil crack network decreased. These results indicate that the increase of PAM reduces the shrinkage strain and defects or pores of saline soil. It proves that PAM can stabilize saline soil under the condition of the dry-wet cycle. Therefore, using water-retaining agents to modify CCC is an effective material to prevent CCC from producing desiccated cracks.

Reviewer #2: line 94: you should give an explanation about silty clay, this is the materials that you used in CCC?

Response: Thank you for your comments. I'm sorry to say that we confused some information about in-situ soil and clinoptilolite when determining the engineering classification of soil. It has been corrected in the manuscript. The details are as follows:

According to the Unified Soil Classification System[49], the in-situ soil and clinoptilolite were classified as low plastic clay (CL) and silt with high liquid limit(MH), respectively.

Reviewer #2: line 124 and line 134: add the ASTM standards.

Response: Thank you for your comments. The compaction characteristics and liquid plastic limit of this paper were tested according to the specification[48]. Therefore, the Test methods of soils for highway engineering (JTG 3430-2020) are cited.

Reviewer #2: line 153: what about the size of the LOCK size? What is the humidity in your experiments? Does it reasonably for your experiments?

Response: Thank you for your comments. (1) The diameter of the LOCK & LOCK BOX is 114mm, the height is 55mm, and the usable volume is about 300mL. The box’s information has been added to the manuscript. (2) According to the literature[19, 21, 50], the temperature is an important factor in the filter paper method test, and the humidity only needs to be kept constant. Therefore, the reference [19, 50] in this paper sets the temperature constant at 20 ℃ and constant humidity. (3) The authors think the experiments are reasonable. Because the test method in this paper is derived from reference [19, 50] and this paper only uses the filter paper method to measure the matrix suction in order to characterize the improvement of the water-holding capacity of clinoptilolite/sodium polyacrylate for CCC. The full suction process will be continuously improved in the future.

Reviewer #2: line 153: Is keeping it for 14 days enough?

Response: Thank you for your comments. Because the author did not express this clearly in the process of writing the paper, we are extremely sorry. ASTM D5298 [50] suggests that the water vapour equilibrium state can be reached after standing for 7 days, which is not the case in actual research. Therefore, based on Sun et al. 2011[19], this paper increases the standing time to 14 days and found that 14 days can fully meet the test requirements of this paper during the test. It has been revised in the manuscript. It is as follows:

‘(3) Put it into a sealed container (LOCK & LOCK BOX, which a diameter is 114mm, height is 55mm, and a usable volume is about 300mL) and keep it for 14 days or more at 20 ℃ and constant humidity [19, 21, 50]to ensure that the filter paper is wet to the moisture content balanced with the CCC suction’.

Reviewer #2: line 333: the discussion part is not enough, what about the micromechanism?

Response: Thank you for your suggestion. First of all, the author thinks your suggestion is very valuable. The water-retaining agents in this paper mainly absorb water physically, without chemical reaction. Therefore, this paper only explores the clay pores through a polarizing microscope, without considering the use of more detailed microscopic analysis (such as SEM, XRD, etc.). Now, the author is using an alkali activator to further explore and will conduct a corresponding microanalysis, which will be reflected in future articles.

Reviewer 3 Report

General comments:

This manuscript conducted a series of experimental studies about the water-holding properties of compacted clay cover modified with Clinoptilolite/Sodium. The authors summarized different kinds of recipes and tried to figure out the best material content. The work is meaningful and practical. However, as a high-quality academic paper, major revisions need to be performed before the acceptance of the paper. Detailed comments are listed as follows:

1. General: Please detailed explain the mechanism of applying such test results to real-life cases (better to draw a schematic diagram). Is your material effective in a mine-contaminated site?

2. General: authors should give a symbol list.

3. General: The authors should replace all the “silty clay” as “Compacted Clay Cover (CCC).”

4. General: Is there any engineering backgroud of silty clay studied in this studies. Authors should list some cases.

Specific comments:

5. About the title: The “silty clay” is not the main content of this paper. The authors mentioned it because it is a component material of compacted clay cover. Authors should reconsider their titles.

6. Line 12: Authors studied “optimum moisture content and atterberg limits of silty clay modified by clinoptilolite and Na-PAA, what is the purpose of the tests.

7. Lines 20-21: should give the quantitative description.

8. Lines 25-38: more information should focus on the mine-contaminated site rather than mineral products. Authors should list the leading mine-contaminated site that may use the CCC methods.

9. Lines 41-52: authors should list the practical engineering cases about CCC cracks and make a specific description (better to make a table).

10. Lines 53-79: All the contents are water-retaining agents. The authors should explain the relationship between the cracks and water-retaining. It is better to add some descriptions of engineering approaches to crack inhibition.

11. Figure 2: the contaminated soil will pollute the surrounding environment (such as underground water, river, and so on). Authors should supplement it with figures.

12. Figure 3: This figure should be modified to refer to other similar figures in other researchers’ articles.

13. Table 1: why not test the OPC and MDD of clinoptilolite? The authors should explain it.

14. About materials: better to list all the photos of materials.

15. Table 2: authors list some cases in this table; however, they should explain their considerations based on which factors.

16. Line 141: authors only use the filter paper tests. The whole SWCC curve needs some other test methods. Authors should explain the reasons that they only use filter paper tests.

17. Line 164: Please correct “30μm thick, 4-6cm2”  as “30 μm thick, 4-6 cm2”.

18. Figure 4, 9, 10: adjust the text size of figures.

19. Lines 210-220: what is the relationship between the Atterberg limits and water-holding capacity? The authors should explain.

20. Figure 7: adjust the text size of figures.

21. Line 278: why do you discuss the fitting model? Is there some relationship with the optimization of the material? Authors should give specific reasons.

22. Line 318: authors should compare their results with others.

23. Figure 9(b): the biomass polymer seems perfect. It looks much better than your materials.

24. Figure 9(a): something wrong with this figure. Check it.

25. Figure 10: guar gum and gellan gum rules differ from others. The authors should explain the reason.

Author Response

Reviewer #3: This manuscript conducted a series of experimental studies about the water-holding properties of compacted clay cover modified with Clinoptilolite/Sodium. The authors summarized different kinds of recipes and tried to figure out the best material content. The work is meaningful and practical. However, as a high-quality academic paper, major revisions need to be performed before the acceptance of the paper. Detailed comments are listed as follows:

Response to Reviewer #3:

Above all, we would like to thank you for the specific and constructive comments on this manuscript. These have helped us reexamine our manuscript critically and make improvements. The point-to-point responses to the reviewer’s comments are listed as follows.

Reviewer #3: General: Please detailed explain the mechanism of applying such test results to real-life cases (better to draw a schematic diagram). Is your material effective in a mine-contaminated site?

Response: Thank you for your advice. The mechanism has been added in Figure 2. It is as follows:

The materials are effective in a mine-contaminated site. Because clinoptilolite has good adsorption performance, it can adsorb heavy metal ions in AMD, improve the breakthrough time of AMD, and improve the chemical compatibility of clay [15-16]. In addition, sodium polyacrylate is a good super absorbent with excellent water retention performance[18, 27-28]. In addition, sodium polyacrylate can block pores [18, 27-28] after water absorption and expansion, thus improving the impermeability of CCC. Under the condition of alternation of dry and wet conditions, sodium polyacrylate can have a self-healing effect because of its characteristics such as water absorption and expansion [18].

Figure 2. Problems to be solved and research approach.

Reviewer #3: General: authors should give a symbol list.

Response: Thank you for your insightful comments. It has been added in the manuscript, which is as follows:

Nomenclature

Na-PAA         Sodium polyacrylate

SWCC           Soil-water characteristic curve

AEV              Air entry value

AEVd             Dimensionless parameter of air entry value

AMD             Acid mine drainage

CCL               Compacted clay cover

WL                                  Liquid Limit

wLd                                 Dimensionless parameter of liquid limit

WP                   Plasticity index

Ip                   Plasticity index

wopt                Optimum moisture content

ρdmax              Maximum dry density

woptd              Dimensionless parameter of optimum moisture content

CL                 Clay with low liquid limit

MH                Silt with a high liquid limit

G                   Replacement content of clinoptilolite

P                   Content of Na-PAA

SSR               Sum of residual squares

Reviewer #3: General: The authors should replace all the “silty clay” as “Compacted Clay Cover (CCC).”

Response: Thank you for your insightful comments. It has been revised in the manuscript.

Reviewer #3: General: Is there any engineering backgroud of silty clay studied in this studies. Authors should list some cases.

Response: Thank you for your comments. I'm sorry to say that we confused some information about in-situ soil and clinoptilolite when determining the engineering classification of soil. It has been corrected in the manuscript. The details are as follows:

According to the Unified Soil Classification System[49], the in-situ soil and clinoptilolite were classified as low plastic clay (CL) and silt with high liquid limit(MH), respectively.

Reviewer #3: About the title: The “silty clay” is not the main content of this paper. The authors mentioned it because it is a component material of compacted clay cover. Authors should reconsider their titles.

Response: Thank you for your advice. The title has been changed to “water-holding properties of clinoptilolite/sodium polyacrylate modified compacted clay cover of tailing pond”

Reviewer #3: Line 12: Authors studied “optimum moisture content and atterberg limits of silty clay modified by clinoptilolite and Na-PAA, what is the purpose of the tests.

Response: Thank you for your comment. Authors think that the changes in optimum moisture content and Atterberg limits are of significance to the water-holding properties. Because with the increase of optimum moisture content and Atterberg limits, more water can be absorbed and retained. Relevant conclusions can be found in reference [40].

Reviewer #3: Lines 20-21: should give the quantitative description.

Response: Thank you for your advice. However, due to the limitation of the polarizing microscope, the pore size can only be analyzed qualitatively. In the future, we will optimize and upgrade the polarizing microscope to better quantitatively analyze the pore size.

Reviewer #3: Lines 25-38: more information should focus on the mine-contaminated site rather than mineral products. Authors should list the leading mine-contaminated site that may use the CCC methods.

Response: Thank you for your advice. Authors think that the development of mineral products is an important reason for mining. Therefore, the development of mineral products is briefly introduced, through which the hazards of mining are introduced, such as AMD. In addition, the sources and hazards of AMD are introduced in detail and summarized in Figure 1 to facilitate others understanding more clearly.

Reviewer #3: Lines 41-52: authors should list the practical engineering cases about CCC cracks and make a specific description (better to make a table).

Response: Thank you for your advice. It has been added in the manuscript, which is as follows:

Fly ash, geosynthetic clay liner (GCL), compacted clay cover (CCC), sludge, and sawdust are commonly used as the source control methods for AMD pollution problems in engineering[2]. The CCC has the advantages of wide source, simple construction, and excellent waterproof and air-blocking performance[5-6], which can effectively inhibit the oxygen and rainwater infiltration into tailing ponds and the reaction of metal sulfide in tailings to generate AMD. Due to the influence of water fluctuation and temperature change, the engineering soil(eg. CCC, Roadbed clay, Foundation clay, and so on.) is easy to crack and loses its stability, thus causing serious secondary environmental disasters (as shown in Table 1). The desiccation cracking width of the CCC can reach 10.0 mm and the depth can become 0.3 m, which can continuously increase to more than 1.0 m if water continues to evaporate and drain away[8-10]. The hydraulic conductivity also increases by about 1-4 orders of magnitude due to the desiccation cracking[11]. Due to the water/gas migration domination channel[7] formed by desiccation cracking, the oxidation rate and quantity of tailings are increased. Therefore, it is necessary to enhance the water retention performance of CCC to prevent desiccated cracks caused by the rapid evaporation of water.

Table 1. Environmental disasters caused by soil cracking or instability

No.

Date (yr)

Location

Harmfulness

Pictures

1

2010

Southwest China

Severe drought caused soil cracking of 1200 dams in Chongqing

2

2022

Laixi, China

There are gullies and cracks in the dam body of the tailings pond, causing no harm

Lack of information

3

2007

Yueyang, China

The mountain on the left bank of the tailings pond has many cracks, threatening the safe operation of the tailings pond

Lack of information

4

2008

Shanxi, China

The tailings pond had been liquefied in a large area and the dam body is unstable, resulting in 277 deaths, 4 missing and 33 injured.

5

2022

Shanxi, China

Collapse accident caused by leakage and instability of tailings pond dam

6

2000

the Philippines

218 dead and 100 missing

7

2005

Indonesia

61 people dead, 90 people were missing and 75 houses were destroyed

Reviewer #3: Lines 53-79: All the contents are water-retaining agents. The authors should explain the relationship between the cracks and water-retaining. It is better to add some descriptions of engineering approaches to crack inhibition.

Response: Thank you for your advice. It has been revised in the manuscript, which is as follows:

Water-retaining agents can quickly absorb and preserve water and reduce desiccation cracking [5]. Soltani et al. [62] show that after adding polyacrylamide(PAM) into the rubber-modified expansive soil, the link between rubber and soil particles is enhanced due to the role of polymer binder. It can further inhibit the crack development of rubber-improved soil. Salemi et al. [18] used sodium polyacrylate to study the self-healing ability and anti-wet/dry cycle ability of geosynthetic clay liners. The results show that the self-healing ability of geosynthetic clay liners is improved by super-absorbent polymer inclusion. The wet/dry cycle test results show that the wet/dry cycle resistance of geosynthetic clay liners can be significantly improved by partially replacing bentonite with a super absorbent polymer. And Zhang et al. [63] showed that with the increase in PAM content, the intersection point and crack length of the compacted saline soil crack network decreased. These results indicate that the increase of PAM reduces the shrinkage strain and defects or pores of saline soil. It proves that PAM can stabilize saline soil under the condition of the dry-wet cycle. Therefore, using water-retaining agents to modify CCC is an effective material to prevent CCC from producing desiccated cracks.

Reviewer #3: Figure 2: the contaminated soil will pollute the surrounding environment (such as underground water, river, and so on). Authors should supplement it with figures.

Response: Thank you for your advice. It has been revised in the Figure 2 of the manuscript.

Figure 2. Problems to be solved and research approach.

Reviewer #3: Figure 3: This figure should be modified to refer to other similar figures in other researchers’ articles.

Response: Thank you for your advice. It has been revised in the manuscript, which is as follows:

Figure 3. Sampling site for in-situ clay.

Reviewer #3: Table 1: why not test the OPC and MDD of clinoptilolite? The authors should explain it.

Response: Thank you for your comment. The reason why not test the OPC and MDD of clinoptilolite is as follows:

The compaction characteristics of clinoptilolite are not the main research goal of this paper, and the impact of clinoptilolite on the compaction characteristics of soil has been studied more. Clinoptilolite will reduce the maximum dry density and increase the optimal moisture content of the soil. This is because the specific gravity of clinoptilolite is smaller than that of soil[22], and the specific gravity of mixed soil decreases, thus the maximum dry density decreases. Because of the special structural characteristics of clinoptilolite [14], it can absorb more water. The increase in the replacement amount of clinoptilolite reduces the particle size distribution of the mixed soil, effectively increasing the optimal water content of the mixed soil [23]. Similar results are obtained in this paper

Reviewer #3: About materials: better to list all the photos of materials.

Response: Thank you for your advice. The photos of materials have been added to the manuscript, which is as follows:

Figure 4. picture of in-situ clay

Figure 5. picture of clinoptilolite

Figure 6. picture of sodium polyacrylate

Reviewer #3: Table 2: authors list some cases in this table; however, they should explain their considerations based on which factors.

Response: Thank you for your comment. The replacement content of clinoptilolite is set based on the economy and the scope of searching for the optimal dosage, and the test results in [55-57] are referred to in the setting process. The following instructions have been added to the manuscript:

In order to explore the influence of clinoptilolite content on soil water-holding capacity, and in consideration of the hydraulic conductivity, adsorption performance, gas resistance and other requirements of CCC, the replacement content of clinoptilolite is set as 0%, 3%, 5%, 10% and 15% according to the literature [55-57].

The content of sodium polyacrylate in this paper is to verify whether it can promote the water-holding capacity of clinoptilolite-modified soil. Therefore, based on the content range of super-absorbent polymers given in reference [51-54], the content of sodium polyacrylate in this test is determined to be 0.35%, which ensures economy and full play. The following instructions have been added to the manuscript:

According to the content range of super-absorbent polymers given in the literature [51-54], the content of Na-PAA used in this paper is determined to be 0.35%, which can avoid the waste of manpower and materials to the greatest extent in the preliminary discussion, and achieve the due modification effect.

Reviewer #3: Line 141: authors only use the filter paper tests. The whole SWCC curve needs some other test methods. Authors should explain the reasons that they only use filter paper tests.

Response: Thank you for your comment. The test method in this paper is derived from references [19, 50]. And in this paper, the improvement of the water-holding capacity of clinoptilolite/sodium polyacrylate for CCC can be characterized by measuring the matrix suction using only the filter paper method. The full suction process will be continuously improved in the future.

Reviewer #3: Line 164: Please correct “30μm thick, 4-6cm2”  as “30 μm thick, 4-6 cm2”.

Response: Thank you for your advice. It has been revised in the manuscript.

Reviewer #3: Figure 4, 9, 10: adjust the text size of figures.

Response: Thank you for your advice. It has been revised in the manuscript.

Reviewer #3: Lines 210-220: what is the relationship between the Atterberg limits and water-holding capacity? The authors should explain.

Response: Thank you for your advice. It has been added in the manuscript, which is as follows: The ability of fine-grained soil to absorb bound water can be directly reflected by liquid limit or plastic limit (i.e. consistency index) [40]. Therefore, the increase of liquid limit or plastic limit indicates that the adsorption capacity of soil to water is improved, which further indicates that the water retention capacity of soil is improved.

Reviewer #3: Figure 7: adjust the text size of figures.

Response: Thank you for your advice. It has been revised in the manuscript.

Reviewer #3: Line 278: why do you discuss the fitting model? Is there some relationship with the optimization of the material? Authors should give specific reasons.

Response: Thank you for your comment. It has been revised in the manuscript, which is as follows:

3.3.2.1. Determine the fitting model

At present, many soil-water characteristic(SWCC) models have been developed [29, 45-46], but SWCC is affected by soil structure, dry density, particle size distribution, stress state and other factors. A previous study[58] has shown that the influencing factors of SWCC are various, and the existing SWCC models may not be directly applied or suitable for certain soil. Therefore, this paper conducts a fitting analysis of several common models and compares the reliability of the model through the sum of residual squares(SSR)[32, 59].

Where =weighting factor, which is equal to 1.0[60]; =measured moisture content at a certain pressure level; and =calculated moisture content from each model at the same pressure level.

In order to further quantitatively analyze the influence of clinoptilolite and Na-PAA on the SWCC of CCC, the Lsqcurve function built in MATLAB is used to fit the SWCC. Four SWCC models are considered during fitting, namely Van Genuchten model [45], the Fredlund & Xing model [29] and the Gardner model [46]. The specific expressions and parameter meanings of each model are shown in table 3. Among them, the  in the FX1 is not the actual residual suction value. Fredlund et al. [29] mentioned that in most cases  can be between 1500 kPa and 3000 kPa, and good fitting results can be obtained for soil-water characteristic curves of different types of soil. When the residual suction value is difficult to determine, it is recommended that  be taken as 3000 kPa [30]. For the convenience of calculation, 3000 kPa is taken in this paper.

Table 3. Soil-water characteristic curve models [29, 45-46]

Models

Equation

Parameters

Van Genuchten model(VG)

: Volumetric moisture content

: Residual moisture content

: Saturated moisture content

s: Matrix suction, kPa

a、n、m: Fitting parameters, a is related to air entry value(AEV),kPa, and m=1-1/n

Fredlund & Xing model (FX)

Fredlund & Xing model (FX1)

: Suction corresponding to the residual moisture content, kPa;

The rest of the parameters as above

Gardner model (GD)

Parameter meaning as above

Table 4 shows the SSR of different SWCC models. As can be seen from table 4, Z0P0 and Z10P0 had the smallest SSR when fitted using the VG model, whereas other samples had the smallest SSR when fitted using the FX1 model. However, the difference between the SRR of the VG model and the FX1 model is very small. And both the two model’s SSR are less than 10-3, indicating that the model provides an appropriate and acceptable fit to the measured data[60-61], so FX1 model fitting results are selected for analysis.

Table 4. Sum of squares of fitting residuals of different SWCC models.

Models

Group No.

Z0P0

Z3P0

Z5P0

Z10P0

Z15P0

Z3P0.35

Z5P0.35

Z10P0.35

Z15P0.35

VG

1.5E-5#

8.4E-4

6.3E-4

3.1E-4#

9.3E-4

1.7E-3

8.8E-4

8.7E-4

5.1E-4

FX

3.9E-4

8.3E-4

5.5E-4

8.0E-4

1.1E-3

2.1E-2

6.7E-4

6.0E-4

5.4E-4

FX1

1.9E-4

3.1E-4#

1.1E-4#

3.3E-4

4.2E-4#

2.7E-4#

2.6E-4#

2.0E-4#

1.2E-4#

GD

7.5E-5

1.1E-3

9.9E-4

4.3E-4

1.2E-3

2.2E-3

1.3E-3

1.4E-3

7.9E-4

Note: # represents the minimum of the SSR.

Reviewer #3: Line 318: authors should compare their results with others.

Response: Thank you for your advice. It has been added to the manuscript, which is as follows:

It can be seen from the pictures that the CCC particles are a flocculation structure, resulting in large and many pores (the pores are occupied by 502 glue), while the particles of clinoptilolite/Na-PAA modified CCC is relatively dispersed, which may be because Na-PAA swells and forming a gel that clogs the clay pores [18, 27-28], or because Na-PAA has strong viscosity after absorbing water, thus forming a "glue"-like effect between clay particles. This may be because, for the unmodified soil, the interaction between soil particles is dominant, and the low viscosity of pore fluid will not inhibit the interaction between particles, so a flocculation structure will be formed. As the viscosity of pore fluid increases, hydrophilic and charged hydrogels are formed. The higher the hydrophilic gel, the better the water-holding capacity of the soil[64]. In addition, clinoptilolite used in this paper is a fine particle [15-16], which can fill pores as much as possible, so the pore size is reduced. Similarly, some researchers obtained similar conclusions[35]. They believe that the flocculation structure of the unmodified soil will form larger pores between particle aggregates, leading to the loss of water in the sample. After adding biopolymer, the sample will form a dispersed structure, making the particles closely connected, thus reducing the loss of water and increasing the water holding capacity. Other results[34] also show that the water-holding capacity of soil is affected by the size of pores. Due to the existence of macropores, the soil will begin to drain under low matrix suction, so its air intake value is low.

Reviewer #3: Figure 9(b): the biomass polymer seems perfect. It looks much better than your materials.

Response: Thank you for your comment. The authors have made an economic analysis of the biomass polymer used in the literature [43] and the polymer used in this paper, and the price is higher than that in this paper. However, the maximum performance improvement is 33.38% compared with this paper, so it can be considered that the Na-PAA used in this article has more economic value than the locust bean gum used in the literature [43]. The supplementary contents are as follows:

As can be seen from Figure 13(a), woptd shows a less pronounced decreasing trend with increasing biomass polymers[43]. The woptd of soil mass modified by clinoptilolite, straw ash, polyacrylamide and Na-PAA showed an increasing trend, indicating that the soil mass modified by these water-retaining agents had a good water-holding capacity. Clinoptilolite and Na-PAA used in this paper have better woptd performance than other water-retaining agents. From Figure 13(b), it can be seen that the wLd of soil increases with the addition of water-retaining agents, while diatomite has a certain inhibition effect on wLd [24], this may be because diatomite and soil water absorption capacity is the same, so the woptd did not significantly improve. Biopolymers [43] have the highest wLd lifting capacity, showing an opposite rule to that of woptd, because biopolymers increase the pore fluid viscosity of soils, and the clay-polymer link network is formed by cationic bridge and hydrogen bond, so the wLd is increased. Although the effect of the Na-PAA used in this paper on the increase of soil liquid limit is not as good as that of locust bean gum in literature [43], the economic benefits of the Na-PAA used in this paper is lower than that of locust bean gum (as shown in Table 7). Clinoptilolite and Na-PAA used in this paper also have good wLd enhancement effects and are better than biochar, attapulgite and straw ash. Therefore, clinoptilolite and Na-PAA have a good lifting effect on wLd and woptd, and show that the CCC modified by both of them has good water-holding capacity and economy.

  Figure 13. The influence of different water retaining agent materials on the optimum water content and liquid limit. (a) Optimum water content woptd, (b) liquid limit wLd.

No.

Materials

Purity specification

Price(/100g)

Contents(refer to dry soil)

Mass of dry soil

Cost

1

Locust bean gum

Food-grade

$ 2.8

eg. 1%

eg.

10kg

$ 2.800

2

Straw ash

-

$ 0.14

eg. 20%

$ 2.800

3

Polyacrylamide

AR

$ 1.66

eg. 0.6%

$ 0.996

4

Attapulgite

-

$ 0.83

eg. 10%

$ 8.3

5

Biochar

Rice straw

$ 0.28

eg. 15%

$ 4.2

6

clinoptilolite

-

$ 0.23

15%

$ 3.45

7

Na-PAA

AR

$ 1.38

0.35%

$ 0.483

Table 7. Comparison of economic benefits of partial water-retaining agents

Reviewer #3: Figure 9(a): something wrong with this figure. Check it.

Response: Thank you for your comment. The data in figure 9(a) is right, after checking. And authors have put the original data in the supplementary materials

Reviewer #3: Figure 10: guar gum and gellan gum rules differ from others. The authors should explain the reason.

Response: Thank you for your comment. The reason why guar gum modify clay’s AEV is higher than this study is that The pore size of guar gum improved soil is similar to that of the original soil, but its cohesion makes AEV larger than that of the original soil. However, the content used was greater than that of Na-PAA in this paper. Therefore, it can be considered that with the increase of Na PAA content in this paper, the increased value of AEV will be greater than that in the literature [35]. It had been explained in the original text as follows:

In this study, AEVd tended to increase with the addition of Na-PAA, while in the literature [35] AEVd decreased after modification with Xanthan gum and gellan gum, while ad increased after modification with guar gum. The porosity of soil after treatment with xanthan gum and gellan gum may be larger than that before treatment, but guar gum increases ad due to its adhesiveness [43]. However, the content of cross-linked polymer Na-PAA in this paper is only 0.35%, so it can be predicted that Na-PAA has better water-holding capacity than guar gum.

Round 2

Reviewer 3 Report

All of my questions have been answered, and the quality of the manuscript has been significantly improved. Therefore, I recommend publishing this paper.